# Simultaneous Temperature and Strain Measurements Using Polarization-Maintaining Few-Mode Bragg Gratings

**DOI:** 10.3390/s19235221

**Published:** 2019-11-28

**Authors:** Chongxi Wang, Zhanhua Huang, Guifang Li, Shan Zhang, Jian Zhao, Ningbo Zhao, Huaiyu Cai, Yinxin Zhang

**Affiliations:** 1Key Laboratory of Opto-electronics Information Technology, Ministry of Education, Tianjin University, Tianjin 300072, China; zhanhua@tju.edu.cn (Z.H.); xyzs10@163.com (S.Z.); enzhaojian@tju.edu.cn (J.Z.); nbzhao@tju.edu.cn (N.Z.); hycai@tju.edu.cn (H.C.); yinxin@tju.edu.cn (Y.Z.); 2CREOL, The College of Optics and Photonics, University of Central Florida, Orlando, FL 32816, USA; li@creol.ucf.edu

**Keywords:** fiber optics sensors, few-mode fibers, polarization-maintaining fibers, fiber Bragg gratings

## Abstract

Simultaneous measurement of temperature and strain was demonstrated using a polarization-maintaining few-mode Bragg grating (PM-FMF-FBG) based on the wavelength and phase modulation of the even $LP_{11}$ mode. The wavelength shift sensitivity and the interrogated phase sensitivity of the temperature and strain were measured to be 10 pm·°C^−1^ and 0.73 pm·με^−1^ and −3.2 × 10^−2^ rad·°C^−1^ and 4 × 10^−4^ rad·με^−1^, respectively, with a discrimination efficiency of 98%. The polarization interference led to selective polarization excitation of the reflection spectra, and the calculated phase sensitivity agreed with the experimental results.

## 1. Introduction

The cross-sensitivity of strain sensors to temperature has been a long-standing issue in optical fiber sensing [1,2,3]. Among the various types of fiber-based strain sensors, the fiber Bragg gratings (FBGs) have become the preferred technology for many applications due to its compactness and the accuracy of its spectral response in the presence of external perturbations. Consequently, considerable attention has been devoted to simultaneous temperature and strain measurement using these sensors.

In principle, the dual parameter techniques [4] developed for simultaneous measurements with conventional fiber interferometers may be adapted to Bragg gratings. The first demonstration of this technique utilized 850 and 1300 nm FBGs for recording measurements in the same region of fiber [5]. Various alternatives that avoid the need to work at widely different wavelength regions involve recording with gratings on either side of a splice between fibers of different diameters [6], using a single long-period grating (LPG) [7], using superimposed FBG/LPG [8], using two different types of FBGs [9,10] and using a tilted FBG [11]. Unfortunately, none of these schemes resulted in particularly well-conditioned interrogating equations. Since the advent of photonic crystal fibers (PCFs), temperature-insensitive strain sensors [12,13] and highly temperature-sensitive PCF sensors with liquid ethanol filling [14] have been reported, providing another method for discriminating temperature. Few-mode fibers (FMFs) have been recently developed in communications applications for their capability to overcome the capacity limit of single-mode fibers (SMFs) [15] and few-mode FBGs (FMF-FBGs) have attracted much research interest to realize mode conversion [16,17,18]. Tilted FMF-FBGs [19] have been proposed to realize the mode conversion from the fundamental mode to the higher-order mode with high conversion efficiency; however, FMFs may also provide new solutions for temperature cross-sensitivity problems [20]. Li et al. demonstrated the simultaneous measurement of temperature and strain in FMFs by utilizing intermodal interference or Brillouin scattering [21]. Guo et al. measured the wavelength shift response of temperature, strain, and bending in FMF-FBGs [22] and found a relatively minor wavelength shift sensitivity difference between the different peaks in reflection spectrum; thus, in practice, it is impossible to distinguish temperature. Zhao et al. showed the resonance depth of the excited high-order core modes can be maximum with FBGs inscribed in few-mode fibers at a tilt angle of 1.4° and tilt angle larger than 2° would excite strong cladding modes [23]. On the other hand, polarization-maintaining FMFs (PM-FMFs) have almost all of the properties of polarization-maintaining single-mode fibers (PM-SMFs) and traditional FMFs. Thus, an FBG inscribed in a PM-FMF could exhibit more abundant features, including temperature discrimination.

In this paper, an FBG was successfully written in a panda-type silica-germanium PM-FMF at a tilt angle of 1°. Mode coupling with selective polarization excitation was analyzed with the reflection spectra. For studying temperature–strain sensitivity in the PM-FMF-FBG, we also used an FMF-FBG inscribed by the same method and a polarization-maintaining single-mode Bragg grating (PM-SMF-FBG) for comparison. We experimentally discovered that the intensity of the reflected light varies sinusoidally with temperature and strain in the PM-FMF-FBG, which has not been observed in the FMF-FBG or PM-SMF-FBG. The phase interrogated from the intensity varied linearly with temperature and strain. Thus, a new method for the complete discrimination of temperature and strain based on wavelength and phase modulation utilizing only one piece of panda-type PM-FMF-FBG was proposed and demonstrated theoretically and experimentally.

## 2. Experimental Setup and Principle

The fiber used in this study was a panda-type silica-germanium PM-FMF (Fiberhome Networks) that supported the $LP_{01}$ and $LP_{11}$ mode groups. The physical parameters of the PM-FMF are shown in Figure 1b, the cladding diameter was 109.45 μm, the core diameter was 16.31 μm and the birefringence $B∼5\times 10^{-4}$. The fibers were hydrogen-loaded in a sealed chamber with a pressure of 10 MPa at 80 °C for one week. The fabrication procedure is depicted schematically in Figure 1c. When inscribing the tilted FBG, the fiber whose coating was stripped was held behind the phase mask. A KrF excimer laser (Coherent BraggStar) operating at 248 nm was used as the UV source. A phase mask was mounted on a displacement stage whose inclination angle can be adjusted. A cylindrical lens was used to focus the UV beam. Decreasing UV laser intensity across the fiber core due to absorption and the tilt angle of 1° lead to the formation of an asymmetric index profile in the grating. This asymmetric profile contributed to a non-zero overlap integral between various spatial modes that would not otherwise be coupled. Hence, cross-couplings were excited, and new reflection wavelengths were produced. Note that the normal FBG without tilt angle inscribed in the PM-FMF would not produce a reflection of high-order modes. After the tilted FBG was fabricated, it was annealed in a thermostat at the temperature of 85 °C for 24 h. The length of the FBG was fixed at 15 mm, and the grating period was fixed at 550 nm. For comparison, a section of few-mode Bragg grating (FMF-FBG), which was inscribed in the same way, and a standard PM-SMF-FBG were tested with the PM-FMF-FBG.

The reflection spectra of the prepared PM-FMF-FBG were measured using an amplified spontaneous emission (ASE) source and an optical spectrum analyzer (OSA). Figure 1a shows the experimental design for the multi-parameter sensing system. Broadband light was passed through an isolator to the ASE to remove any back-reflection influence. The light was then divided into two orthogonal linearly polarized components by a polarization beam splitter (PBS). A fiber circulator was used to pick up one of the linearly polarized components, while the other component was terminated. The state-of-polarization of the circulated polarized light was adjusted by a polarization controller. To keep the selected polarization in the polarization controller, the PM-FMF-FBG was spliced with a PM-SMF without any core offset. Surely, splicing with core offset will increase the coupling of high order modes, but will increase the loss. To test the effect of tensile stress, we placed the fiber on two fiber holders mounted on opposing translation stages. By keeping one end fixed and moving the other end, we were able to apply stress on the fiber. The initial length of the fiber subjected to the tension was 20 cm, and the FBG was centered along the fiber length. We increased the length by 5 μm at a time, which corresponded to a strain value of 250 με. Meanwhile, the temperature was changed from 25 to 135 °C in 5 °C increments using a heating panel.

The relationship between the two counter-propagating waves in the produced PM-FMF-FBG can be given by
(1)$$\beta_{\mu }^{+}+\beta_{v}^{-}=\frac{2\pi \cos(\theta )}{\Lambda }$$
where $\beta_{\mu }^{+}$ and $\beta_{\nu }^{-}$ are the propagation constants of the two counter-propagating waves, Λ is the grating period, θ is the tilt angle and μ,ν∈{01x,01y,11ox,11oy,11ex,11ey}, where x and y indicate orthogonal polarizations, and o and e represent odd and even modes, respectively.

For the scenario of self-coupling, both counter-propagating waves share the same mode (μ=ν); hence, Equation (Equation 1) can be simplified as
(2)$$\lambda_{\mu }=2n_{eff,\mu }\Lambda /\cos\left(\theta \right)$$
where $\lambda_{\mu }$ is the resonant wavelength, θ is the tilt angle and $n_{eff,\mu }$ is the effective refractive index of the corresponding $LP_{\mu }$ spatial mode. The relationship between the excited resonant wavelength and the two cross-coupled spatial modes ($LP_{\mu }^{+}\to LP_{v}^{-}$) is given by
(3)$$\lambda_{\mu ,v}=\left[n_{eff,\mu }+n_{eff,v}\right]\Lambda /\cos\left(\theta \right)$$
where μ,v∈{01x,01y,11ox,11oy,11ex,11ey}, μ≠v.

The measured reflection spectra of the PM-FMF-FBG are shown in Figure 2a. As the state-of-polarization of the input light was adjusted using the polarization controller, the reflected light varied gradually from x-polarization (red) to y-polarization (blue). Generally, the reflection spectrum would be the superposition of the two lines. The wavelength separation Δλ, shown in Figure 2a for the polarized modes of $LP_{01}$, $LP_{11e}$, and $LP_{11o}$, was 0.47 nm, 0.51 nm, and 0.52 nm, respectively. We can calculate the values for the birefringence from the measured wavelength separation of $LP_{01}$, $LP_{11e}$, and $LP_{11o}$ polarized modes as 4.3 × 10^−4^, 4.6 × 10^−4^ and 4.7 × 10^−4^, respectively, which agreed with the specified birefringence $B∼5.0\times 10^{-4}$ measured with the original fiber without inscription. The resonant wavelength of each mode conversion can be theoretically estimated using Equations (Equation 2) and (Equation 3), as illustrated in Figure 2b. Note that the cross-coupling of $LP_{\mu }^{+}\to LP_{v}^{-}$ and $LP_{v}^{+}\to LP_{\mu }^{-}$ share a reflection peak and similar coupling characteristics at the same wavelength. For simplicity, the notation $LP_{\mu }\to LP_{v}$ is employed to represent cross-coupling between the two modes. The mode intensity profile analysis was conducted with a tunable laser source (TLS) and a CCD camera. The insets in Figure 2a show the mode profiles of the reflected beam at different peak wavelengths. The $LP_{01}$ mode profile was observed at (iv) and the $LP_{11}$ mode profile was observed at (i) and (iii).

The intensity of the reflection peaks is mainly determined by the fraction of mode power excited into the FBG at the joint of PM-SMF and PM-FMF, the tilt angle of the FBG and the state-of-polarization of the input light. In this special device, even $LP_{11}$ mode was mainly excited and reflected by the tilted FBG and the highest reflection was achieved at peak (iii), which corresponds to the self-coupling $LP_{11ex}↔LP_{11ex}$. The reflection peak was centered at the wavelength of $\lambda_{\mu }=$ 1534.46 nm with the full-width at half-maximum (FWHM) ~0.2 nm. The other three resonant wavelengths, namely (i), (ii), and (iv), correspond to (1531.32 nm), (1533.95 nm), and (1536.63 nm), respectively.

Derived from Equation (Equation 3) with $\theta =0^{\circ }$, the dependence of the Bragg wavelength on temperature and strain can be calculated as [24]
(4)$$\Delta \lambda =\lambda \left[\left(\alpha +\frac{1}{n_{eff}}\cdot \frac{dn_{eff}}{dT}\right)\Delta T+\left(1-P_{e}\right)\Delta \varepsilon \right]$$
where α is the thermal expansion coefficient, $n_{eff}$ is the effective refractive index of each mode, and $P_{e}$ is the strain-optic coefficient.

From the physical parameters of fused silica in [24], simply using the measured values of $\alpha =0.52\times 10^{-6}$, $dn_{eff}/dT=1.1\times 10^{-5}$, $n_{eff}\approx 1.46$, $P_{e}=0.348$ in Equation (Equation 4), the wavelength shift sensitivity of temperature and strain was calculated as 12.4 pm·°C^−1^ and 1.0 pm·με^−1^, respectively.

As shown in Figure 2a, the coupled-mode theory for LP modes [25] can explain the majority of the peaks in the reflection spectrum more easily than that for its vector modes. However, vector modes are required for studying the important effects of the polarization properties of fields [26].

In an interesting phenomenon shown in Figure 2a, both the x- and y-polarized light are reflected at peaks (ii) and (iii). For example, when only x-polarized light is excited into the fiber, some y-polarized light would be reflected at the wavelength of peak (iii). This phenomenon would result in polarization mode interference at the splice point of PM-SMF and PM-FMF. However, the reflected light at peak (i) and peak (iv) is very clean, which would theoretically result in much weaker interference.

The polarization-interference intensity can be written as
(5)$$I=I_{0}+I\cos\left(\Phi \right)$$
where Φ=2πLB/λ is the phase difference between the two reflected polarization modes, *L* is the interference length of ~10 cm, and $B=n_{x}-n_{y}$ is the birefringence between the slow axis and fast axis of the modes in PM-FMFs. The phase Φ is temperature and strain-dependent, and can be calculated as
(6)$$\begin{array}{c} \Phi =2\pi \mathrm{LB}/\lambda \\ \frac{\Delta \Phi }{\Delta T}=\frac{2\pi }{\lambda }\cdot \frac{\Delta (\mathrm{LB})}{\Delta T}=\frac{2\pi }{\lambda }\cdot \frac{\Delta L\cdot B+\Delta B\cdot L}{\Delta T}=\frac{2\pi LB}{\lambda }\left(\alpha +\frac{\Delta B}{B\Delta T}\right)\\ \frac{\Delta \Phi }{\Delta \varepsilon }=\frac{2\pi }{\lambda }\cdot \frac{\Delta (\mathrm{LB})}{\Delta \varepsilon }=\frac{2\pi }{\lambda }\cdot \frac{\Delta L\cdot B+\Delta B\cdot L}{\Delta \varepsilon }=\frac{2\pi LB}{\lambda }\left(10^{-6}+\frac{\Delta B}{B\Delta \varepsilon }\right) \end{array}.$$

In our previous work [27], we measured the spectrum resonance of temperature and strain based on polarization interference using the phase-mask method. Using this method, the temperature and strain sensitivity can be expressed as
(7)$$\begin{array}{c} \Phi =2\pi \mathrm{LB}/\lambda =m\pi ,\\ \frac{\Delta \lambda }{\lambda \Delta T}=\left(\frac{\Delta L}{L\Delta T}+\frac{\Delta B}{B\Delta T}\right)=\alpha +\frac{\Delta B}{B\Delta T}\\ \frac{\Delta \lambda }{\lambda \Delta \varepsilon }=\left(\frac{\Delta L}{L\Delta \varepsilon }+\frac{\Delta B}{B\Delta \varepsilon }\right)=10^{-6}+\frac{\Delta B}{B\Delta T} \end{array}$$
where *m* is a positive integer.

Based on the temperature and stain sensitivity of $LP_{11}$ mode, measured as –0.2172 nm·°C^−1^ and 0.0057 nm·με^−1^ [27], respectively, and Equation (Equation 7), we find
(8)$$\left\{\begin{array}{c} \frac{\Delta B}{B\Delta T}=-1.4\times 10^{-4}\\ \frac{\Delta B}{B\Delta \varepsilon }=2.7\times 10^{-6} \end{array}\right..$$

From the physical parameters of fused silica in [24], simply using the measured values of $\alpha =0.52\times 10^{-6}$, L=10 cm, and $B=5.0\times 10^{-4}$, the phase sensitivity for temperature and strain of −2.9 × 10^−2^ rad·°C^−1^ and 7.6 × 10^−4^ rad·με^−1^, respectively, may be calculated from Equation (Equation 6) and Equation (Equation 8).

To simultaneously recover temperature and strain, the majority of reported methods utilize so-called dual-parameter techniques as follows. Two different optical signals $S_{1}$ and $S_{2}$ are derived from a single sensor. These signals are related to the temperature *T* and strain ε by the matrix equation
(9)$$\left(\begin{array}{c} S_{1}\\ S_{2} \end{array}\right)=\left(\begin{array}{cc} a & b\\ c & d \end{array}\right)\left(\begin{array}{c} T\\ \varepsilon \end{array}\right),$$
where a(c) and b(d) are the strain and the temperature sensitivities, respectively. The solution to Equation (Equation 9) exists if
(10)D=ad−bc≠0.

To determine the effectiveness of this approach for temperature and strain discrimination, we introduce the normalized parameter E proposed by Triolet et al. [28] to quantify the coupling effect of strain and temperature:(11)$$E=\left|D\right|\cdot {\left[\left(a^{2}+c^{2}\right)\cdot \left(b^{2}+d^{2}\right)\right]}^{-1/2}.$$

## 3. Experimental Results

Figure 3a shows the change in the reflection spectra with the temperature changing from 35 to 135 °C in 15 °C increments, and Figure 3b shows the change in the spectra with the strain changing from 0 to 1750 με in 250 με increments. The peak wavelength changed with both temperature and strain. Moreover, the intensity of peak (iii) showed a large change with temperature and strain.

The wavelength shift response to temperature and strain are as illustrated in Figure 4a,b, respectively. The response of the wavelength shift has fairly good linearity and exhibits similar sensitivities for all considered peaks. The temperature sensitivities of the four peaks were measured to be 10 pm·°C^−1^, agreed with the calculated values of 12.4 pm·°C^−1^. The strain sensitivity of peak (i) was measured to be 0.71 pm·με^−1^, while that of the other three peaks was 0.73 pm·με^−1^, agreed with the calculated values of 1.0 pm·με^−1^. Hence, we cannot discriminate the cross-sensitivity of temperature and strain using the wavelength shift of multiple peaks without producing a large error in the temperature or strain. To simultaneously measure temperature and strain, another parameter, such as the phase modulation, should be used.

Figure 5 shows the peak intensity response to temperature ranging from 35 to 135 °C under different strains. From Figure 5a, we can see that the intensity variation of peak (i) was relatively small with an amplitude of ~0.3 nW, suggesting weak interference. Figure 5b shows that the intensity response of peak (ii) is generally linear between 60 °C and 110 °C, which could be useful for sensing, but the amplitude is relatively small. Figure 5c shows that the reflected intensity of peak (iii) increases with temperature below ~60 °C and then decreases, with an amplitude of ~20 nW, and the extinction ratio is larger than 10 dB. Finally, the approximately linear intensity response to temperature from 35 to 100 °C of peak (iv) is shown in Figure 5d.

Based on the results shown in Figure 5, we have two choices for simultaneously measuring temperature and strain. We could utilize the intensity response of peak (ii); however, this choice has drawbacks: a limited temperature range of ~50 °C and a small sensitivity. The other option is to utilize peak (iii), which has a wide temperature measurement range of 100 °C from its peak to trough, corresponding wavelength shift of 1 nm.

To confirm the dependence of the temperature in peak (iii), we increased the measured temperature to 185 °C. The intensity response of peak (iii) is illustrated in Figure 6. As we can see from the red curve, the reflected intensity began to increase at the temperature of ~150 °C, suggesting a sine-wave-like dependence. We also changed the polarization state of the input light via the polarization controller. As we can see from the blue curve, the temperature corresponding to the max intensity remains unchanged at 56 °C, but the amplitude of the curve decreases due to the decreased input intensity of x-polarized light. Moreover, the intensity increases when the temperature is larger than 160 °C for x and y polarized light.

We also compared the temperature response of PM-SMF-FBG and FMF-FBG to the PM-FMF-FBG. The FMF-FBG was inscribed using the same method as that for PM-FMF-FBG, and the inset of Figure 7 shows the FMF-FBG reflection spectrum. As shown in Figure 7, the peak intensity of the standard PM-SMF-FBG did not vary as the temperature changed. Likewise, Figure 7b shows that the intensity of reflected light for the FMF-FBG did not vary significantly as the temperature changed. On the other hand, the intensity response of peak (iii) in PM-FMF-FBG, shown in Figure 7a, was perfectly waveform fitted with a determination coefficient $R^{2}=0.9991$.

The intensity response to the strain of peak (iii) was examined experimentally, as illustrated in Figure 8a. We can see the strain sensitivity was negative with temperature ranging from 35 to 55 °C and positive with temperature ranging from 65 to 135 °C. The calculated slope at different temperatures is shown in Figure 8b. The fitted curve is a sinusoidal function, which suggests that the reflected intensity of peak (iii) also varies sinusoidally with strain.

Figure 9 shows the response of the reflected intensity of peak (iii) with the simultaneous variation of temperature and strain. The fitted response of peak intensity is
(12)$$P=10.827+8.819\cos\left(-3.2\times 10^{-2}T+4\times 10^{-4}S+1.4918\right)$$
where *P* is the peak intensity, *T* is the temperature, and *S* is the strain.

From Equation (Equation 12), we can interrogate the phase from the detected intensity of peak (iii) by
(13)$$\Phi =\arccos\left(\frac{P-10.827}{8.819}\right)-\pi$$
noting that Equation (Equation 13) can only give the phase of temperature from about 55 °C to 130 °C for this special device. To interrogate the phase of a full range of the temperature, other methods must be used, which will be discussed below.

When the minimum detectable intensity of the photodiode is fixed, the maximum reflected intensity of peak (iii) determines the phase resolution of the sensor. In our experiment, the overall output power of the broadband source was set to 10 mW, and at wavelength 1538 nm, the intensity was measured to be ~10 μW. The PBS and fusion splicing between PM-SMF and PM-FMF also introduced a large loss. In practice, 1 mW laser diodes with the linewidth of 5 nm are enough for the sensor, and will largely improve the phase resolution. Optimization of the tilt angle of the FBG and fusion splicing can also improve the phase resolution.

Applying the interrogation Equation (Equation 13) for this special device, the interrogated phase change induced by temperature and strain is plotted in Figure 10. As shown in Figure 10a, the phase varied linearly with the temperature and the R square value was 0.979. The phase sensitivity to temperature was measured to be −3.2 × 10^−2^ rad·°C^−1^, agreed with the calculated values of −2.9 × 10^−2^ rad·°C^−1^. Around 50 °C corresponding to the phase of about 0 rad, the result calculated from interrogation Equation (Equation 13) shows a relatively large error. Moreover, at the left and right side of 0 rad phase point, the temperature response exhibited the same measured intensity in this special device (e.g., 30 °C, 70 °C), thus, the phase could not be calculated directly. We determined the sign of the phase by the intensity change of applied strain. As shown in Figure 8a, if the reflected intensity increases with increasing strain, then we can tell that the temperature is located above 50 °C and the sign of the phase is negative. In production, we should avoid the 0 rad point locating in the measuring range of the sensor. This can be realized by adding a phase bias to the interference, for example, packaging the sensor with a preset strain using piezoelectric ceramics driven by a signal generator. Properly reducing the length of the PM-FMF can also increase the measuring range of the sensor. As the devices may have different initial phase, calibration of the devices is an important process. Likewise, Figure 10b shows that the phase varied linearly to strain, with the R square value of 0.987. The phase sensitivity to strain was measured to be 4 × 10^−4^ rad·με^−1^, agreed with the calculated values of 7.6 × 10^−4^ rad·με^−1^.

Combining the wavelength shift and the interrogated phase change of peak (iii), simultaneous measurement of temperature and strain can be obtained as follows: (14)$$\left(\begin{array}{c} \Delta \lambda \\ \Delta \Phi \end{array}\right)=\left(\begin{array}{cc} 1.0\times 10^{-2} & 7.3\times 10^{-4}\\ -3.2\times 10^{-2} & 4\times 10^{-4} \end{array}\right)\left(\begin{array}{c} \Delta T\\ \Delta S \end{array}\right)$$

From Equation (Equation 14), the opposite sign holds for the phase change, which is different from the wavelength shift. Using Equation (Equation 11), a comparison of the discrimination efficiency for different methods is calculated in Table 1. As we can see, the discrimination efficiency of our work can be calculated as 98%, which is better than most methods proposed in the literature.

In the experimental apparatus, a strain parallel to the fiber axis was given to the sensor. However, in pretty much all real-world applications bend-like deformations would induce an error in the measurements. Ulrich et al. pointed out that bending-induced birefringence is essentially a stress effect and gave the following equation [31]
(15)$$\Delta \beta =0.25k_{0}n^{3}\left(p_{11}-p_{12}\right)\left(1+\nu \right)r^{2}/R^{2}$$
where $B=\frac{\Delta \beta }{k_{0}}$ is the bending induced birefringence, $k_{0}=\frac{2\pi }{\lambda_{0}}$, $\lambda_{0}$ is wavelength in vacuum, *n* is the refractive index of fiber, $p_{11}$ and $p_{12}$ are strain-optical coefficients, ν is Poisson’s ratio, *r* is the fiber radius, R is the bending radius. With the measured phase sensitivity to temperature of −3.2 × 10^−2^ rad·°C^−1^, to keep the temperature error less than 0.01 °C, the bending induced phase change should be less than 3.2 × 10^−4^. Inserting into Equation (Equation 15) the published [32] material constants of fused silica [$n=1.46;\nu =0.17;(p_{1}1-p_{1}2)=-0.15$] and with $\lambda_{0}=1538\;\mathrm{nm}$, r=62.5μm, fiber length L=10cm, we can calculate that the bending radius *R* should be larger than 0.83 m. Moreover, Zhao et al. [23] showed that bending would have an impact on the reflected intensity and wavelength shift of $LP_{11}$ mode on condition that the bending radius was less than 0.2 m, which will cause error in the process of phase and wavelength interrogation. So, in practice, the sensor should be carefully packaged.

## 4. Conclusions

We investigated the mode-coupling properties of PM-FMF-FBGs with selective polarization excitation, and the temperature and strain sensing properties of the PM-FMF-FBGs were fully explored. The wavelength shift sensitivities of temperature measured at four Bragg wavelengths were found to have the same value of 10 pm·°C^−1^, agreed with the calculated values of 12.4 pm·°C^−1^. Likewise, the wavelength shift sensitivity of strain at the Bragg wavelength (i) was measured to be 0.71 pm·με^−1^, while the wavelength shift sensitivity at the other three wavelengths was 0.73 pm·με^−1^, agreed with the calculated values of 1.0 pm·με^−1^. A new phenomenon in which the intensity of reflected light varied sinusoidally with temperature or strain was experimentally observed; however, this phenomenon was not found in PM-SMF-FBG or the inscribed FMF-FBG. Within a strain change of 1750 με, the reflected intensity variation was relatively small; thus, the variation could be linearly approximated at certain temperatures. Below 56 °C, the slope of the reflection intensity response to strain was positive and that from 56 to 156 °C was negative. A phase interrogation equation was given by fitting the measured dataset using the SQP optimization algorithm and the interrogated phase from the reflected intensity of peak (iii) varied linearly with temperature and strain. The interrogated phase sensitivity of temperature and strain were measured to be −3.2 × 10^−2^ rad·°C^−1^ and 4 × 10^−4^ rad·με^−1^, respectively, agreed with the calculated values of −2.9 × 10^−2^ rad·°C^−1^ and 7.6 × 10^−4^ rad·με^−1^, respectively. The discrimination efficiency of our method was calculated to be 98%. By analyzing the reflection spectra with selective polarization excitation, a theoretical explanation for the temperature and strain sensitivity was proposed based on polarization interference, and the calculated sensitivity agrees well with the experimental results. The proposed method simplifies interrogation systems by requiring one wavelength shift detection rather than dual-wavelength detection. In practice, the sensor should be carefully packaged, as the sensor is based on the coupling between modes and polarization interference, a variation on the crosstalk between modes and extra coupling due to curvatures, twisting, etc. is traduced in a variation of the output intensities, modifying the measured phase. In addition to being used as a temperature and strain sensor, PM-FMF-FBGs could also be used as an intensity tunable FBG reflector with a tunable range larger than 10 dB.

## Figures and Tables

**Figure 1 sensors-19-05221-f001:**
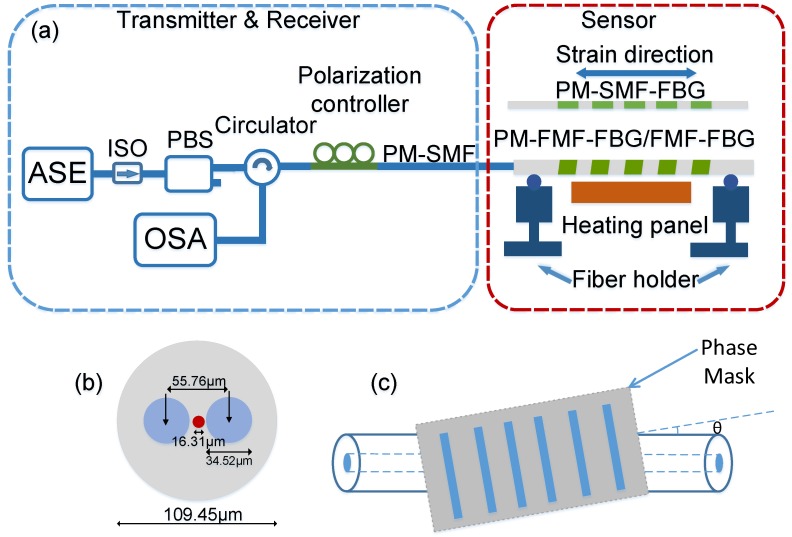
Experimental scheme (**a**) The proposed temperature and strain sensing system. (**b**) Geometry parameters of the polarization-maintaining few-mode fibers (PM-FMF). (**c**) Fabrication procedure for the tilted fiber Bragg grating (FBG).

**Figure 2 sensors-19-05221-f002:**
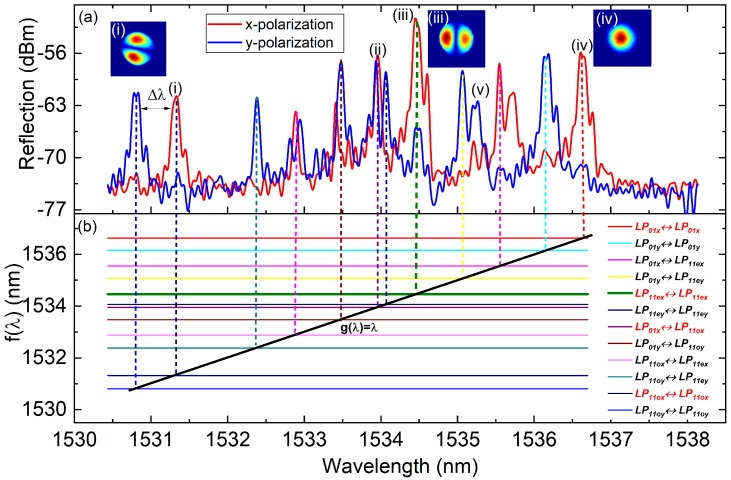
Reflection spectra. (**a**) Reflection spectra of the two-mode polarization-maintain few-mode Bragg grating (PM-FMF-FBG) under x-polarization excitation (red) and y-polarization excitation (blue). Insets show the beam profiles at the wavelengths of interest. (**b**) Theoretical estimation of the resonant wavelengths of corresponding mode couplings, $f\left(\lambda \right)=\left[n_{\mu }\left(\lambda \right)+n_{\nu }\left(\lambda \right)\right]\Lambda /\cos\left(\theta \right)$ or $f\left(\lambda \right)=2n_{\mu }\Lambda /\cos\left(\theta \right)$ (colorful lines) and g(λ)=λ (black line).

**Figure 3 sensors-19-05221-f003:**
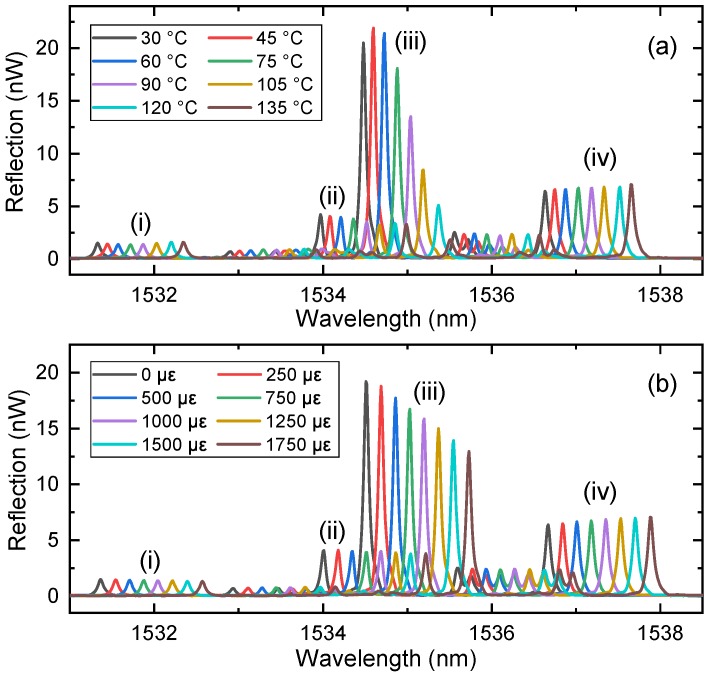
Spectrum response of temperature and strain. (**a**) Response of the filtered reflection spectra of the PM-FMF-FBG to temperature. (**b**) Response of the filtered reflection spectra of the PM-FMF-FBG to strain.

**Figure 4 sensors-19-05221-f004:**
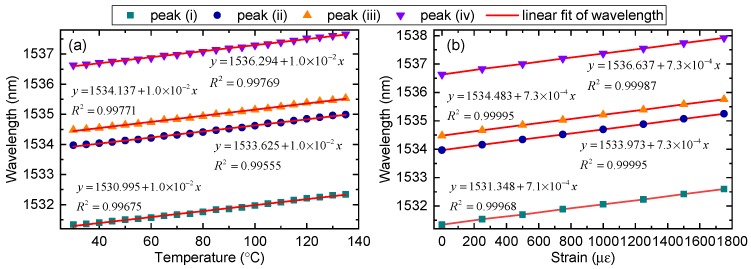
Wavelength shift of peaks. (**a**) Temperature response of the PM-FMF-FBG. (**b**) Strain response of the PM-FMF-FBG.

**Figure 5 sensors-19-05221-f005:**
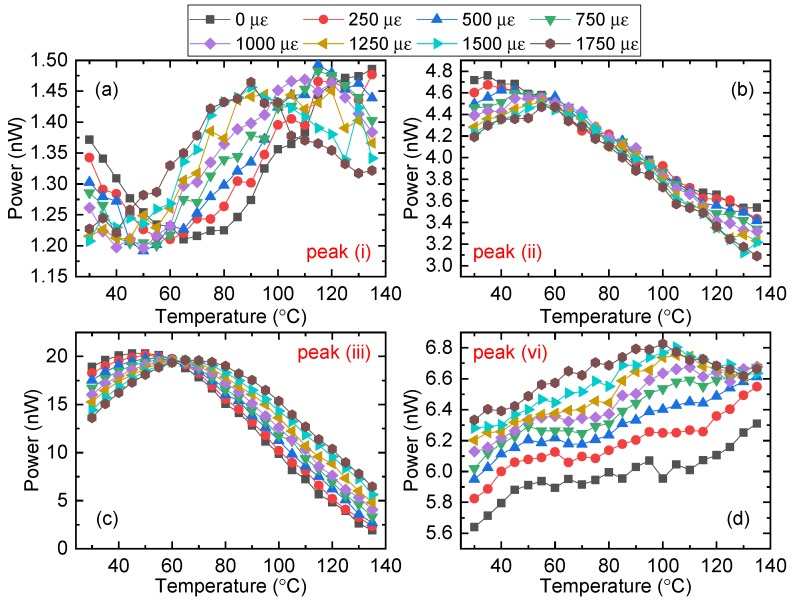
Intensity changes of peaks (**a**–**d**) Temperature responses of peaks (i)–(iv), respectively, under different strains.

**Figure 6 sensors-19-05221-f006:**
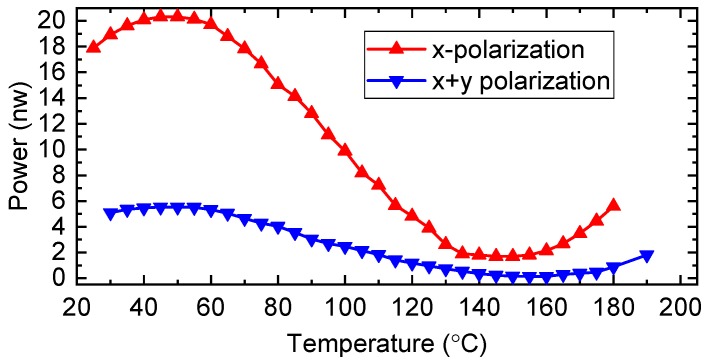
Temperature response of peak (iii) under x-polarization excitation (red) and both x- and y-polarization excitation (blue).

**Figure 7 sensors-19-05221-f007:**
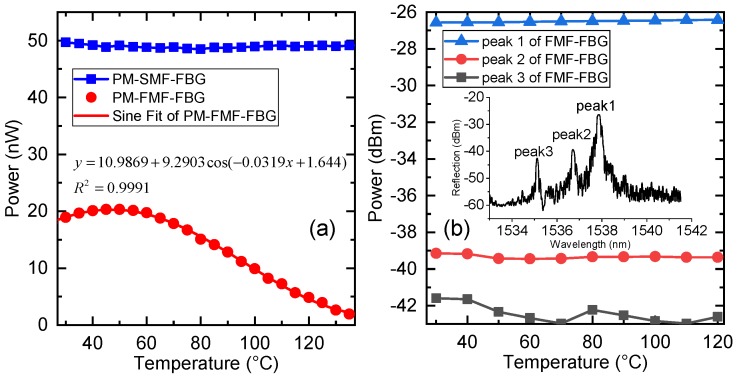
(**a**) Temperature response of peak (iii) in PM-FMF-FBG and the corresponding sine fitted curve (red) and PM-SMF-FBG (blue). (**b**) Temperature responses of the three peaks of the inscribed FMF-FBG. The inset shows the FMF-FBG reflection spectra.

**Figure 8 sensors-19-05221-f008:**
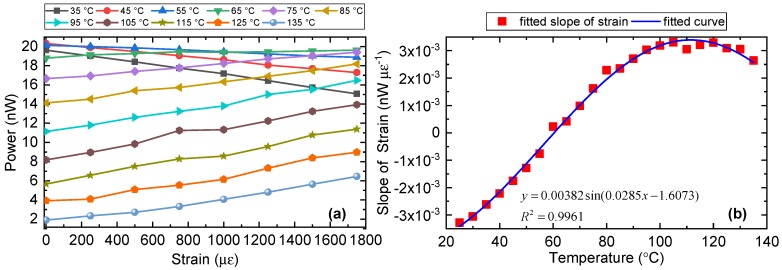
(**a**) Strain responses of the PM-FMF-FBG under different temperatures. (**b**) Calculated slope of the curve (strain sensitivity) in (**a**) and the corresponding sine fitted curve.

**Figure 9 sensors-19-05221-f009:**
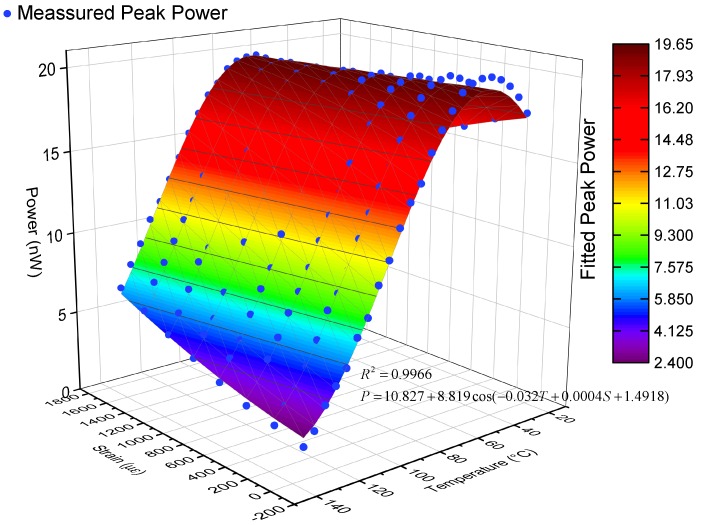
Intensity response of peak (iii) in PM-FMF-FBG with simultaneous temperature and strain variation and the corresponding sine fitted curve determined using the sequential quadratic programming (SQP) optimization algorithm.

**Figure 10 sensors-19-05221-f010:**
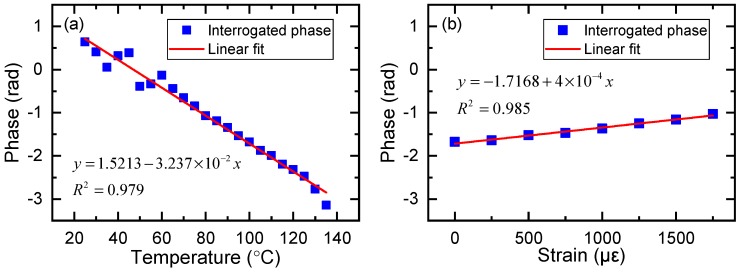
Phase modulation of peak (iii)(**a**) Interrogated phase response of peak (iii) in PM-FMF-FBG to temperature (at 0 με). (**b**) Interrogated phase response of peak (iii) in PM-FMF-FBG to strain (at 100 °C).

**Table 1 sensors-19-05221-t001:** Comparison of the discrimination efficiency for different methods proposed in the literature.

Discrimination Technique	Discrimination Efficiency E	Reference
LPG alone	35%	[7]
FBG in capillary tube	70%	[29]
FBG in few-mode POF	6%	[30]
Superimposed FBG/FBG	7%	[11]
Superimposed FBG/LPG	99%	[28]
PM-FMF-FBG alone	98%	This work

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
