# Peer review of "Simultaneous Temperature and Strain Measurements Using Polarization-Maintaining Few-Mode Bragg Gratings"

_sensors, 2019, doi:10.3390/s19235221_

Round 1

Reviewer 1 Report

The authors proposed a method for measuring temperature and strain using PM-FMF-FBG. The method is novel, but I believe that the sensitivity of the experimental parameters is not particularly prominent. For temperature response, a sinusoidal change trend has been discovered, which is not as good as linearity in sensing. So, I believe that the draft may not be suitable for publication on Sensors.

The author needs to examine the grammar and capitalization of the article carefully. I found a lot of cases where the first word was lowercase. In Figure 2, some reflection peaks have prominent high reflectivity, this phenomenon should be explained. There are too many lines in Figure 3 and looks particularly messy. The trend of sinusoidal changes for temperature response, how does the author distinguish the temperature? For example, when the output power is 15 nW in Figure 5 peak (3) and the temperature corresponds to ~35 degrees Celsius and ~90 degrees Celsius, how does the author distinguish between the two temperatures? According to these results, the PM-FMF-FBG seems not a good candidate for sensing application.

Author Response

Dear Reviewer:

We sincerely thank you for your careful and hard work of our manuscript (ID: 625018, title: Simultaneous temperature and strain measurements using polarization-maintaining few-mode Bragg gratings). Your thoughtful comments and constructive suggestions improve the quality of our manuscript a lot. We have made revisions for every comment in the revised manuscript as the locations stated in every response below. We also highlighted all the changes in response to the reviewers’ comments.

Thank you again for all your time and hard work on this paper.

Best wishes,

Reviewer 2 Report

The temperature-strain cross sensitivity of Fiber Bragg Gratings is a classic issue of sensing applications. In this manuscript the authors claim that these two contributions to the FBG wavelenght shift can be disentangled by using a tilted polarization-maintaining few-mode-fiber fiber-bragg-grating (PM-FMF-FBG). The device presents several Bragg resonances, due to the cross-excitation of the allowed spatial modes operated by the tilted FBG. The presented technique is based on acquiring simultaneously the intensity and the wavelength shift of one of these resonances. The mentioned Bragg resonance responds to the strain with a wavelength shift (and almost no intensity change) and to the temperature with both a wavelength shift and an intensity variation. Such intensity variation is sinusoidal, is maximized for selective polarization excitation, and only appears in PM-FMF-FBG.

I found the manuscript very well written and organized. In this regard i just suggest to check for typos (i detected two in lines 95 and 133) and to enlarge Fig.2. As for the content, I believe there are two points that demand a more thorough discussion, as they are crucial for the reproducibility of the results and the possibility to employ the described sensor in real applications. In particular:

The intensity variation of peak (iii) with temperature stems from a modal interference: the relative phase between the interfering modes changes with temperature, causing a sinusoidal I(T) behaviour. But who decides the initial phase, (say, the phase at 20 degrees) of this interference? Is it always the same in different devices? Is it controllable? This should definitely be clarified.

In the experimental apparatus a strain parallel to the fiber axis is given to the sensor. However, in pretty much all real-world applications FBG sensors are also subject to orthogonal components (think for example of bend-like deformations). Would the described behavior of the PM-FMF-FBG still hold in this situation? Given the nature of the modal interference it seems to me this is not something that can be just assumed. I strongly suggest the authors to discuss this point too.

Author Response

(The authors gave the same response as above.)

Reviewer 3 Report

The authors present a theoretical and experimental study on PM-FMF-FBGs. The mode-coupling properties with selective polarization excitation, and the temperature and strain sensing properties of the PM-FMF-FBGs were fully explored, and a new phenomenon in which the intensity of reflected light varied sinusoidally with temperature or strain was experimentally observed. The introduction provides sufficient background, the results are clearly presented and supports the conclusions well. So I think that this manuscript could be accepted in present form.

Author Response

Dear Reviewer:

We sincerely thank you for your careful and hard work of our manuscript (ID: 625018, title: Simultaneous temperature and strain measurements using polarization-maintaining few-mode Bragg gratings). Your thoughtful comments and constructive suggestions improve the quality of our manuscript a lot. We have made revisions for every comment in the revised manuscript as the locations stated in every response below. We also highlighted all the changes in response to the reviewers’ comments.

Thank you again for all your time and hard work on this paper.

Best wishes,

Chongxi Wang

Reviewer 4 Report

The authors presented a polarization-maintaining few-mode Bragg grating sensor for simultaneous measurements of temperature and strain. LP11 mode is used, of which the intensity varies sinusoidally with temperature and strain. The authors showed the experimental data for temperature and strain measurements. The presented work is organized well. However, the results show the proposed technique may have problems. There are some comments:

The authors claim that large tilt angle of the grating can excite strong cladding modes. Is the used angle optimized or not? Why the authors choose a tilt angle of 1˚? The unit of dBm is used to devote the intensity in Figure 3 while the unit of nW is used in Figure 5. The reviewer thinks it is better to keep it same. Actually, the coefficient is about -0.2 nW/˚C (60~130˚C), which is not acceptable. The result will be limited by the resolution of the instrument. Hence, it is important to improve the sensitivity.

Author Response

(The authors gave the same response as above.)

Round 2

Reviewer 1 Report

I believe my concerns about the sensing scheme have been adressed, and satisfactory revisions have been made according to the comments.

Reviewer 4 Report

Thanks the authors for answering the reviewer's concerns. This manuscript can be acceptable in the present form.